# A Systematic Review on Contamination of Marine Species by Chromium and Zinc: Effects on Animal Health and Risk to Consumer Health

**DOI:** 10.3390/jox15040121

**Published:** 2025-08-01

**Authors:** Alexandre Mendes Ramos-Filho, Paloma de Almeida Rodrigues, Adriano Teixeira de Oliveira, Carlos Adam Conte-Junior

**Affiliations:** 1Center for Food Analysis (NAL), Technological Development Support Laboratory (LADETEC), Federal University of Rio de Janeiro (UFRJ), Cidade Universitária, Rio de Janeiro 21941-598, RJ, Brazil; alexandremrf11@gmail.com (A.M.R.-F.); adriano.oliveira@ifam.edu.br (A.T.d.O.); conte@iq.ufrj.br (C.A.C.-J.); 2Laboratory of Advanced Analysis in Biochemistry and Molecular Biology (LAABBM), Department of Biochemistry, Federal University of Rio de Janeiro (UFRJ), Cidade Universitária, Rio de Janeiro 21941-909, RJ, Brazil; 3Graduate Program in Veterinary Hygiene (PPGHV), Faculty of Veterinary Medicine, Fluminense Federal University (UFF), Vital Brazil Filho, Niterói 24220-000, RJ, Brazil; 4Analytical and Molecular Laboratorial Center (CLAn), Institute of Chemistry (IQ), Federal University of Rio de Janeiro (UFRJ), Cidade Universitária, Rio de Janeiro 21941-909, RJ, Brazil; 5Center for Studies of Invertebrates and Vertebrates of the Amazon (NEIVA), Federal Institute of Education, Science and Technology of Amazonas (IFAM), Manaus Centro Campus (CMC), Manaus 69020-120, AM, Brazil; 6Graduate Program in Sanitary Surveillance (PPGVS), National Institute of Health Quality Control (INCQS), Oswaldo Cruz Foundation (FIOCRUZ), Rio de Janeiro 21040-900, RJ, Brazil; 7Graduate Program in Food Science (PPGCAL), Institute of Chemistry (IQ), Federal University of Rio de Janeiro (UFRJ), Cidade Universitária, Rio de Janeiro 21941-909, RJ, Brazil; 8Graduate Program in Chemistry (PGQu), Institute of Chemistry (IQ), Federal University of Rio de Janeiro (UFRJ), Cidade Universitária, Rio de Janeiro 21941-909, RJ, Brazil; 9Graduate Program in Biochemistry (PPGBq), Institute of Chemistry (IQ), Federal University of Rio de Janeiro (UFRJ), Cidade Universitária, Rio de Janeiro 21941-909, RJ, Brazil

**Keywords:** fish, chromium, zinc, seafood, health risk, risk assessment

## Abstract

Potentially toxic elements, such as chromium (Cr) and zinc (Zn), play essential roles in humans and animals. However, the harmful effects of excessive exposure to these elements through food remain unknown. In this sense, this study aimed to evaluate the anthropogenic contamination of chromium and zinc in aquatic biota and seafood consumers. Based on the PRISMA protocol, 67 articles were selected for this systematic review. The main results point to a wide distribution of these elements, which have familiar emission sources in the aquatic environment, especially in highly industrialized regions. Significant concentrations of both have been reported in different fish species, which sometimes represent a non-carcinogenic risk to consumer health and a carcinogenic risk related to Cr exposure. New studies should be encouraged to fill gaps, such as the characterization of the toxicity of these essential elements through fish consumption, determination of limit concentrations updated by international regulatory institutions, especially for zinc, studies on the influence of abiotic factors on the toxicity and bioavailability of elements in the environment, and those that evaluate the bioaccessibility of these elements in a simulated digestion system when in high concentrations.

## 1. Introduction

Marine environments are constantly exposed to contaminants from various sources, including heavy industry, oil extraction, mining operations, fertilizers, and pesticides, among other practices which may cause pollution [1]. Toxic metals and other materials are typically quantified and analyzed to assess risks to human health and the environment. However, another type of pollutant presents as much of a risk as these: the essential elements of the soil. These elements, such as zinc (Zn) and chromium (Cr), are present in many organisms and are necessary for various essential bioprocesses. Deficiency of these factors causes various adverse effects. However, despite their central role in the function of the majority of species, if there is a concentration above the necessary level of these essential elements, there will be a variety of side effects that, depending on the severity, can cause death [2]. These metals can be found in pesticide residues and industrial effluents from steel mills, textile industries, and antifouling paints, potentially contaminating the aquatic environment, especially through the irregular discharge of effluents from these industries into water [1,2,3,4,5]. When they reach the oceans and large bodies of water, there is a significant risk of seafood contamination, which eventually affects the human population that consumes these animals [3]. Therefore, it is essential to monitor the levels of these metals in the environment to avoid any impact caused by overexposure.

Zinc is essential for most organisms and is involved in several animal metabolic processes. Approximately four grams of metal is present in the human body, with 90% found in the bones and muscles. In addition, it is a component of more than 300 enzymes and numerous other proteins [4]. However, excessive oral intake of zinc can cause adverse effects such as nausea, vomiting, abdominal cramps, epigastric cramps, and diarrhea. In addition to gastrointestinal symptoms, prolonged consumption of toxic levels of zinc can cause copper deficiency and T-cell deactivation and may play an essential role in the development of cancer [4]. This metal can be released into aquatic environments through the discharge of industrial sewage from metallurgical plants and industrial fertilizers [5]. In water, the predominant chemical form is free zinc (Zn^2+^), especially in environments with low organic matter and acidic pH, which is deposited in the sediment for later absorption by biota, where it accumulates mainly in the gills, leading to problems with oxygen absorption [3].

Cr is present in all organisms and, despite its low concentration in the body, has been identified as a key component of insulin action and blood glucose regulation. Furthermore, it is essential to break down and absorb carbohydrates, proteins and fats [5]. The most common naturally occurring chemical species of chromium are Cr (0) and Cr (III). This species is present in bioprocesses and is considered less toxic because it is not permeable to biological membranes [6]. However, Cr (VI) species are more commonly associated with anthropogenic activities, mainly in the electroplating, textile, and construction industries [5]. Most of the Cr present in marine animals is Cr (III), which is natural and considered a micronutrient. However, anthropogenic interference has increased the presence of Cr (VI), which is associated with several pathologies [5]. Both types of Cr, when absorbed, can undergo alkylation and have their oxidative states changed to more reactive types, such as Cr (VI), which is converted into Cr (V) and Cr (IV). Cr (VI) enters cells through general sulfate transporters. This specific form of chromium has several genotoxic effects, which include breaking bonds in double-stranded DNA and causing a decrease in ascorbate and biological thiols, such as glutathione (GSH) [5]. Notably, exposure to Cr (IV) has been associated with different types of cancer. In 1980, the International Agency for Research on Cancer (IARC) identified it as a Group I human carcinogen [7,8].

Considering the side effects of the ingestion of high levels of these two essential elements in the human body and the increasing level of toxic waste containing them being released into the environment, it is necessary to analyze the levels present in the fish products being consumed, as marine animals are affected the most by the elevated levels of effluents. It is worth highlighting that the choice to evaluate these two metals is related to common sources of emission and the lack of information regarding these elements, which are important for health but have toxic potential.

## 2. Materials and Methods

Following four sequential stages, the two authors (A.M.R.F and P.A.R) first independently conducted a preliminary selection of identified abstracts and paper titles. The search was limited to English publications between 2012 and 2025. Editorials, letters, reviews, mini-reviews, and Ph.D. theses were excluded. The results are reported in accordance with the Preferred Reporting Items for Systematic Reviews and Meta-Analyses (PRISMA 2000) [9] statement and summarized in the PRISMA checklist, presented in Appendix A. This systematic review has been registered with the OSF Registries© (https://doi.org/10.17605/OSF.IO/GRXAQ).

Abstracts and papers were removed if they did not investigate associations between animals/matrix (marine species) and the presence of Zinc (Zn) and Chromium (Cr). Thus, four criteria were selected for articles to be excluded from the selection: not studying marine species (Reason 1), or works on the freshwater environment (Reason 2), studies only soil or sediment without correlation with water or animals (Reason 3), works that addressed the results superficially without clear justifications or results that stood out in relation to other works that managed to carry out a more in-depth approach (Reason 4). Some studies considered essential but not found in any research bases were added, such as those that address complementary data on chemical species of the elements studied and their mechanisms in the evaluated organisms.

The data for each article was collected individually by the authors of the work, seeking to extract the same standardized information from each document, which was later extracted into a spreadsheet and checked by all authors in order to confirm the information. The data evaluated in all articles were as follows: study site, species evaluated, metal and its chemical species, element concentration, result of the non-carcinogenic risk assessment and carcinogenic risk (if present), water parameters measured (if evaluated), and changes caused in animal and/or human health.

### 2.1. Focus Question

The question was developed using the population, intervention, comparison, and outcome (PICO) method. The following questions were formulated: What are the primary sources of contamination of the metals studied (Zn and Cr)? Which chemical species of the elements are mainly responsible for the contamination? What risks do Zn and Cr pose to the final consumer of contaminated fish products? Was the presence of contaminants sufficient to alter the observed organisms?

### 2.2. Information Sources

A literature search was performed using Medical Subject Headings (MeSH) terms in the PubMed, Web of Science, and Embase databases. The initial screening process was performed between March and May 2025. Further directed searches were conducted by checking the reference lists of relevant articles. Search Component 1 (SC1)—Population search: “Fish OR crab OR lobster OR muscle OR “swimming crab” OR squid OR “marine species” OR shrimp OR bivalve OR crustacea OR Seafood”. Search Component 2 (SC2)—Intervention search: “Zn OR zinc OR Cr OR chromium”. Search component 3 (SC3): “Biomagnification OR bioaccumulation OR “health risk” OR contamination OR “Risk assessment” OR “human health risk” OR “environmental contamination”. After retrieving the Search Component results, the Boolean operator “AND” was used to combine SC1, SC2, and SC3.

The data obtained were used to produce maps and graphics using QGIS 3.40.3, and additional information on the identified species was obtained from the FishBase platform (version 04/2025). The vulnerability stage of the species was determined based on the IUCN endangered species list.

### 2.3. Risk of Bias Assessment

Possible sources of bias include the inclusion/exclusion criteria, the impact of missing data, and missing primary results. A preliminary search was conducted in multiple databases and languages to minimize selection bias, covering various article types and publication years. This exploratory phase allowed for the identification of knowledge gaps and helped select inclusion and exclusion criteria that aligned with this review’s objectives. Furthermore, all selected studies were assessed for the quality of the data presented, clarity in the discussion of the results, and presentation of a well-described methodology appropriate for the study’s purpose. All studies were carefully evaluated, and those that did not present consistent results, discussions, and methods, when necessary, were excluded or contextualized in the debate to avoid distortions in the risk estimates. An assessment of the bias of the experimental studies explored in this review is presented in Table 1, grouped by justification.

## 3. Results

A total of 1629 articles were identified in PubMed, 1594 in Embase, and 84 in Web of Science, totaling 3307 articles. Of these, 595 were duplicates. After the exclusion of repeated articles, 2712 articles remained. After reading the titles and abstracts, 2362 articles were excluded because they were not aligned with the study objective, leaving 350 articles. After reading the manuscripts in full, 46 studies that met the study’s aim and selection criteria were selected. In the next stage, 21 articles were added due to the complementarity of data on the chemical species of the elements studied, their mechanisms in the evaluated organisms (15), and regulatory information regarding the permitted limit of each element (7). Therefore, 67 studies were included in this review (Figure 1). Among the reasons for excluding articles were those that did not study marine species (Reason 1 = 458), those that studied the freshwater environment (Reason 2 = 107), those that dealt only with soil or sediment without correlation with water or animals (Reason 3 = 277), and works that superficially addressed the results without clear justifications or results that stood out compared to other works that managed to carry out a more in-depth approach (Reason 4 = 1824).

Most of the studies found were related to the determination of multiple elements, including Cr and Zn, but none were directly related to the evaluation of only these two elements, and very few evaluated Cr or Zn in isolation. In general, the articles that studied only these elements were related to toxicity studies or their dynamics in the environment. Only a few studies have focused on a single organism, whereas most have studied various species. The animals analyzed during the literature search were fish, mussels, crabs, shrimp, sharks, gastropods and squid. The observed studies were identified in several parts of the world (Figure 2), especially in estuarine regions with rich biodiversity and breeding grounds for many marine species. The studies were mainly concentrated in Asia, emphasizing populous and highly industrialized countries (China and India), followed by Europe, with studies in areas of lower anthropogenic impact and environmental protection (the Canary Islands). The African continent was also studied, with work in the northern (Algeria) and southern regions (South Africa), and only one study was conducted on the American continent, specifically in the Galapagos Islands, Ecuador, which is also an environmental protection area.

The articles selected to compose this review mainly sought to quantify several elements, including Cr and Zn in commercially important fish species, molluscs, and crustaceans. The concentrations found in each study were compiled into the table below (Table 2).

The articles compiled in this review were also evaluated regarding the trophic niche of each species (Figure 3).

The fish species were classified according to the International Union for Conservation of Nature (IUCN) Red List of Threatened Species. In this classification, species can be divided into not evaluated, insufficient data, least concern, near threatened, vulnerable, endangered, critically endangered, extinct in the wild, and extinct. No data were found regarding the other animal groups covered in this study, only fish (Figure 4).

## 4. Discussion

### 4.1. Origin and Circulation of Cr and Zn in the Marine Environment

The studied metals can be released into the ocean from several sources, with Zn being found mainly in waste from the metallurgical and fertilizer industries and chromium found in waste from the painting industries, industries involved in leather tanning, steel production, textile manufacturing, and as an antifouling agent in marine paints [6]. Although found in various contamination sources, they are abundant and used together in electroplating, phosphate fertilizers, and fossil fuel waste [5]. It is important to emphasize that both Zn and Cr present in antifouling paints are constantly released into the sea due to degradation caused by the interaction of salt water with vessels [10]. Other sources of heavy metals, such as industrial sewage, are of great concern, especially in regions where sewage is released into the aquatic environment without adequate treatment.

Zn is commonly found in the form of ZnO, both in its elemental form and as nanoparticles, and also as Zn omadine [16]. Although the latter is less commonly used, these Zn compounds are easily absorbed by marine animals and humans and usually do not undergo decomposition before being absorbed. In aquatic environments, the predominant chemical species is free zinc (Zn^2+^), especially in environments with low organic matter and pH below 8. pH, salinity, dissolved organic matter, and calcium strongly influence zinc speciation and toxicity in water. Other inorganic Zn species can also be found in aquatic environments, such as ZnCO_3_^0^, ZnSO_4_^0^, ZnOH^+^, Zn(OH)_2_^0^, ZnCl^+^, Zn(Cl)_2_^0^, Zn(Cl)_3_, ZnHPO_4_, and Zn(Cl)_4_^2−^. In marine environments, the ZnCl_n_^2−n^ species becomes more prevalent with increasing salinity, despite the fact that Zn^2+^ remains the predominant species, representing around 30% of environmental zinc [50].

Chromium is generally found as Cr (0) (or metallic Cr), Cr (III) and Cr (VI), or CrO_4_, commonly found in nature and also in the composition of such antifouling paints, and Cr (V) and Cr (IV), found in the environment due exclusively to anthropogenic sources. Once absorbed by organisms, Cr (III) can be converted into Cr (V) and Cr (IV), both highly toxic forms, as they can interact more easily with proteins and DNA molecules [5]. In the environment, Cr (III) and Cr (VI) are the most stable forms. Cr (III) reacts with seawater at pH 8.1, generating mainly ionic compounds, approximately 85% of which are present as Cr(H_2_O)^4^ (OH)_2_^+^, but it can also be observed as CrO_2_^−^, representing around 13.5% [51]. As for Cr (VI) in the average marine pH of 8.1, it is mostly observed as chromate ions (CrO_4_^2−^), with a minority present as HCrO_4_^−^ [51].

Both contaminants can also be found in fossil fuels, generally as metallic forms. These fuels can react with the marine environment to form oxides or sulfites, increasing their capacity to be absorbed by organisms [5]. In addition, all these compounds have a higher density than ocean water, causing them to accumulate in marine sediments, despite the fact that some compounds can also be dissolved in ocean water [51].

Considering the toxic potential of hexavalent chromium species, the literature indicates the presence of concentrations of 1–10 μg L^−1^ in surface waters and 1–2 mg L^−1^ in natural water bodies, with very low concentrations reported in non-industrialized areas and higher concentrations reported in areas of intense industrialization [6], as identified by Mishra and Mohanty [17] in water samples after treatment to make them potable, originally collected near a leather industry in the Kanpur region, India, with values up to 6.2 mg L^−1^ of Cr (IV). However, according to USEPA [52], the Cr (IV) limit in potable water should be up to 0.1 mg L^−1^.

The presence of Zn in aquatic environments, similar to that of Cr, is related to an eutrophic climate. However, Zhao et al. [53] highlighted that high concentrations of Zn in water are detected in areas where *Microcystis aeruginosa* (cyanobacteria) blooms are observed. These microalgae can efficiently absorb and transfer Zn to other trophic levels of the food chain. Among the species that can establish this link is the silver carp (*Hypophthalmichthys molitrix*), which is commonly found in this environment and feeds on planktonic algae. It is worth noting that this fish species, in addition to being consumed by humans, is also used in the biological control of blooms. In this study, the researchers identified that microalgae exacerbated Zn bioaccumulation in silver carp through digestion, absorption, and contact with the skin mucosa [53].

### 4.2. Maximum Cr and Zn Levels and the Response in Aquatic Animals

Regarding the impact of these elements on animal health, Cr enters the body’s cells mainly in the form of chromate ion or CrO_4_^2−^, which has a structure similar to PO_4_^3−^, and occurs through the chloride-phosphate transport channel [54]. The acute effects, such as changes in glycogen and protein levels, as demonstrated by Vutukuru [18] and Krumschnabel and Nawaz [19], Cr also causes other side effects in fish, such as reduced cell viability [40], decreased white blood cell count, red blood cell count, and hemoglobin [20] impact on reproductive cells that can reduce fertility [21] and increased blood clotting time [22]. These alterations began at 0.005 mg of Cr per liter of water. There are also some chronic effects caused by prolonged chromium intake. In vivo, experiments by Arunkumar et al. [23] showed a reduction in the lymphocyte count, a decrease in the production of antibodies, and a decrease in the weight of the spleen, caused by Cr (III) and Cr (VI). Other in vitro experiments have also shown worrying effects, such as erosion of the fin and fin rays [55], reduced embryo survival rate [56], reduced survival rate, and DNA changes [24]. Bakshi and Panigrahi [56] reported that chronic exposure to Cr (VI) can cause reduced larval growth, reduced survival rate, and erosion of fin morphology and fin rays. In addition to inorganic species, CrPic is an organic Cr (III) used in nutritional supplements, which can also have adverse effects on organic function and fish health, such as disorders in lipid metabolism and potential toxicity in the liver and at the transcription level [25]. In fish fingerlings, acute exposure to Cr causes hyperactivity and loss of balance, whereas chronic exposure to Cr causes erratic swimming and convulsions. In addition, a reduced survival rate of fingerlings has also been observed. Organic Cr may be more bioavailable than inorganic species due to its organic ligands, chemical properties, and high biological activity [25,57].

Toxicity studies evaluating the LC50 of Cr (IV) have identified different lethal concentrations depending on the species, the study’s laboratory conditions, and the animal’s age, among other factors. Other studies have shown that at a concentration of 124.23 mg L^−1^, a decrease in the frequency of leukocytes, erythrocytes, heterophils, lymphocytes, monocytes, and eosinophils was observed in peripheral blood samples in pacu [26], and at a concentration of 91.4 mg L^−1^, histopathological changes were identified, such as steatosis or tissue edema and necrosis in the kidneys, liver, skin, and gills [27]. Marianaro et al. [28] in their study evaluating the impact of exposure to 1, 10, and 100 nM Cr (IV) on the reproductive health of mussels *(Mytilus galloprovincialis*), identified the bioaccumulation of the metal in the gonads and morphological damage, mainly when exposed to the highest concentration tested. This study evaluated new pollution biomarkers, such as poly (ADP-ribose) polymerase 1 and 2 (PARP-1 and PARP-2), which are DNA cleavage sensor enzymes that are active in response to metal cytotoxicity to maintain genomic integrity. This study identified that chromium affects sperm chromatin and promotes molecular changes in protamine-like proteins, affecting their binding to DNA in the sperm nucleus during spermiogenesis.

Zinc can cause dangerous side effects when in excess, despite its function in proteins and bioprocesses. Although data on the subject are scarce, some authors have identified that Zn accumulates mainly in the gills, which can lead to problems with oxygen absorption [3] and also affects fish growth and gastrointestinal pathologies [58]. Regarding the development phase, it has been observed to increase the number of deformed embryos and the mortality rate of normal embryos, and in many cases, it increased the time required to hatch. In the larval stage, malformations of the eyes, spine, brachial arches, and jaw were predominantly observed in Zn-exposed fish [58]. McRae et al. [29] in their study with fish exposed to Zn caused increases in catalase activity and lipid peroxidation, significantly inhibited calcium influx, and stimulated sodium influx. Wu et al. [59] evaluated the effects of zinc nanoparticle contamination in *Mytilus edulis*. These Zn nanoparticles, especially zinc oxide (nZnO), are emerging as the third most produced nanomaterial in the world, with applications in solar cells, optoelectronic devices, biomedicine, antibacterial materials, and personal care products. These findings indicate that nZnO uptake exacerbates hypoxia-induced oxidative stress, delaying redox recovery and prolonging oxidative damage during reoxygenation, indicating a prolonged oxidative imbalance. While animals in the control group typically recovered from hypoxia-induced stress, the presence of nZnO in the organism interrupted this process by impairing antioxidant defenses. Zheng et al. [60] evaluated the effect of zinc on zebrafish (*Danio rerio*) during the early stages of life. The article indicates that exposure to a concentration of 10 μM (650 μg/L) during the first five days after fertilization did not lead to changes in body weight or malformations, nor did it affect the survival rate or hatching success of F0 and F1 larvae. At a concentration of 30 μM (2000 μg/L), delayed hatching was observed in the F0 and F1 generations, and significant changes in Zn and Se homeostasis were observed in F0 and F1. Other changes have also been identified in adult fish, such as imbalances in Zn, Se, and Mn homeostasis in different tissues. The study also suggested that fish may carry memories of Zn exposure during the embryonic larval period into adulthood and transmit them to the next generation.

Regarding bioaccumulation and biomagnification, both metals bioaccumulate in biota in the order of preference of gills > liver > skin > muscle for Cr [6] and liver > ≈ intestine > stomach ≈ gills > muscle for Zn [41]. It is worth noting that abiotic factors in water can affect metal bioaccumulation. In this sense, it was observed that water at pH 6.5 provides a maximum amount of Cr (VI) deposited in the gills compared to other internal organs, but at pH 7.8, other organs present greater tropism than the gills, as for Cr (III) due to its higher stability and lower membrane permeability, most Cr (III) that gets ingested is not absorbed, leading to a lower bioaccumulation and biomagnification [6]. Comparatively, organic Cr species have a higher bioaccumulation potential than inorganic species [25]. Another important aspect related to bioaccumulation and the regulation of metal concentrations in the body is the investigation of genes associated with the homeostasis of metal concentrations in the body. Ma and Wang [41] identified, through molecular evaluation, the influence of the Zip gene in the regulation of metal concentrations in conditions of deficiency and excess. In situations where the Zn diet is deficient, the ZIP1 gene is expressed. Its expression favors the production of proteins on the surface of the intestinal epithelium, which are responsible for increasing Zn uptake. There is also a reduction in the expression of the ZnT1 gene, which reduces the efflux of Zn from intestinal cells. In situations where the animal diet is rich in Zn, there is an increase in the expression of the Zip8 gene, responsible for promoting an influx of Zn and negative regulation of genes responsible for increasing Zn absorption, such as Zip1 [41].

Studies have found evidence of both the absence of biomagnification for Cr (IV) and its presence in some fish species, with carnivorous animals that feed on bottom-dwelling species having the highest Cr values [6]. In contrast, Zn has higher concentrations in predators at the top of the chain, which is concerning, given that humans are part of this group [12]. No information was found on the minimum intake of either metal required for animal welfare.

### 4.3. Metal Levels in the Studied Seafood Species

Some international regulatory agencies have determined the maximum permissible concentrations of chromium and zinc in seafood, including the CMLCF (Chinese), which sets a maximum of 2.0 mg kg^−1^ [46], the CFS (Hong Kong), which sets a maximum Cr concentration of 1.0 mg kg^−1^ [48], and the FDA (USA), which sets a maximum of 12 mg kg^−1^ [47]. As for zinc, only the WHO set a limit for its presence of 40 mg kg^−1^ [49], and for maximum limits in water, only the WHO defined clear limits for both, being 40 mg kg^−1^ for Zn and 0.1 mg kg^−1^ for Cr [49]. All limits and the corresponding regulatory agencies are listed in Table 2. It is worth noting that despite the existence of these limits, as these elements are essential to human and animal health, performing fundamental functions, there is a very subtle threshold between the essential and toxic concentrations, depending on the organism, such as aspects of animal/human metabolism, environmental conditions that may interfere, and the chemical species absorbed and its interaction with the organism.

Of the 20 studies listed in Table 2, seven were conducted in environments with minimal anthropogenic activity, with five of the collection sites in the Canary Islands [13,32,34,39,43] and one in the Galapagos Islands [10]. In most studies, Cr values were below the limit, unlike Zn, which exceeded the limit in several studies. Although most studies have been conducted with fish, significant concentrations of both contaminants have been identified in mussels and crabs. The concentration of these contaminants in mussels is due to their role as environmental filters, which leads them to absorb and concentrate large quantities of toxic materials dispersed in the water. In crabs, the concentrations of these elements are related to their feeding habits, as these animals feed on decomposing matter in the ocean floor’s sediments, where metals tend to accumulate [61].

In areas with more human presence and activity, the concentration of metals in animals was even more concerning, with some articles surpassing 1000 mg/kg of Zn and 10 mg kg^−1^ of Cr. The highest concentrations were observed in Lebanon in the Mediterranean Sea, with mean concentrations of 2384 mg kg^−1^ for *Mytilus edulis* (mussel) and 2700 mg kg^−1^ for *Portunus trituberculatus* (crab) [3]. As for chromium, the worst results originated from Thailand [11], with minimum mean values found in *Isognomon ephippium*, 4.9 ± 2.9 mg kg^−1^ of Cr, and reaching its peak in *Thais gradata*, with a mean concentration of 20.5 ± 24.2 mg kg^−1^. Shaheen et al. [42] found up to 392.06 ± 19.22 mg kg^−1^ of zinc in *Corica soborna* species and 1.75 ± 0.12 mg kg^−1^ of chromium in an individual of the same species, with Zn being the highest concentration identified among this animal group.

As previously mentioned, more than half of the studies were conducted on fish, mostly carnivores, followed by omnivores and a minority of herbivores (Figure 3). Correlating this information with the concentrations indicated in Table 2, it was possible to identify some species. *Trachurus trachurus*, for example, presented significant concentrations of Zn and Cr. This species is carnivorous and is considered a predator of fish and crustaceans. Furthermore, in general, high concentrations of both metals were identified in this same study, with only one of the six species being herbivorous and all the others being carnivorous. These data corroborate other findings, which indicate higher concentrations of both metals in top predator carnivores. The high concentration of metals in the Durban region, where the study was conducted, can be justified by intense industrialization and precarious sanitation [33]. Based on the IUCN red list, *Trachurus trachurus* stands out as a species vulnerable to extinction (Figure 4) due to its commercial importance and consequent overfishing. However, high concentrations of Cr have been reported to reduce fertility [21], which may contribute to reducing the fishing stock of the species.

Another finding that should be highlighted is the vulnerability to extinction of *Pseudupeneus prayensis*, studied in the environmental protection region of the Canary Islands, Spain, according to the Red List represented in Figure 4. Although the concentration of metals identified at this location is not higher than the average values detected in the Durban region, monitoring the contamination of species at this stage of conservation is essential, especially regarding the impact of contamination by potentially toxic metals. This species is also carnivorous and feeds on benthic invertebrates. Sediment is the leading deposit site for contaminants in aquatic environments [52], and consuming benthic species is an essential source of these metals. Another species that stands out in terms of its classification on the red list, *Mycteroperca olfax* (carnivorous), has an endangered status, and its contamination study was carried out in the Galápagos environmental protection area, which also raises an alert regarding anthropogenic activity in this region [10].

### 4.4. Effects of Zn Overexposure to Humans

Despite being essential for humans, Zn can have many adverse effects when ingested in abundance. According to the Institute of Medicine Panel on Micronutrients, Washington (DC), the Recommended Dietary Allowance of Zinc for adults is 11 mg/day for men and 8 mg/day for women [62]. Although there are side effects of ingesting values below the recommended threshold of 8 mg Zn per day, like neurological disorders, infertility, skin conditions, immune dysfunction, and growth retardation [4], the effects of ingesting toxic amounts of this element (above 40 mg per day) are numerous and potentially more dangerous.

Overexposure to Zn can cause gastrointestinal effects, which include diarrhea, vomiting, and abdominal cramps [63]. It also impacts the brain, causing lethargy and focal neuronal deficits [4]. However, despite the severity of these symptoms, they are not the most alarming. Zn poisoning can affect the function of all cells and several hundred zinc-dependent enzymes. When present in abundance, it competes with copper for absorption by enterocytes, causing copper deficiency, which leads to further complications such as anemia, cardiac malfunction, imbalance in cholesterol levels, and neurological symptoms [64]. In in vitro studies, excess Zn in cell culture induced monocytes to secrete pro-inflammatory drugs but also inhibited T cell function, which would cause a reduced immune response to infection [64].

### 4.5. Chromium Intoxication in Humans

There are two main stable chemical forms of chromium. Cr (III) is an essential micronutrient important for insulin-dependent glucose metabolism and is found in the environment. In contrast, Cr (VI) is generally related to anthropogenic action and is considered toxic. This difference in the origin of the chemical types of chromium resulted in different Recommended Dietary Allowances, for Cr (III) of 1.5 mg per day and 0.003 mg per day for Cr (VI) according to US EPA [65].

Chromium (III) reacts with DNA and other biomolecules. It can accumulate inside the cell, causing it to respond with several components of the intracellular matrix. However, it is considered less toxic due to its lower mobility and insolubility in water. These characteristics indicate that Cr (III) is only absorbed by the cell via phagocytosis or passive diffusion, making its absorption very slow and challenging to accumulate inside the cell [62,63]. Chromium (IV) has a higher oxidation state and is therefore more mobile, facilitating its ability to cross cell membranes [66], and can penetrate the nucleus as it is bound to histones [5]. Despite its greater ability to cross membranes, it is considered less genotoxic and capable of interacting with fewer biomolecules [6,7,8]. However, the reason it is considered more toxic is that it is reduced to the more reactive Cr (III) and highly reactive intermediate species, Cr (IV) and Cr (V), once inside the cytoplasm and nucleus, if they are previously converted [5].

Reaching the nucleus, Cr (VI) can be reduced by binding to DNA and being converted to Cr (III). The most common reduction pathway involves GSH, which reduces Cr (VI) by removing an electron and generating a chemical form that is more reactive than the metal, Cr (V). However, the same pathway can also remove two electrons at the same time, generating Cr (IV), which is also a highly genotoxic compound. Other conversion pathways can be based on NADPH reductase, but require further elucidation [5].

### 4.6. Consumer Health Risk Assessment

Animals as food are potentially dangerous to human health when they contain large concentrations of contaminants. Especially if the local consumer population has a food culture of consuming fish products with high frequency and/or quantity, some studies carry out a human health risk assessment. This assessment is based on equations such as the estimated daily/weekly/monthly intake (EDI/EWI/EMI) and Hazard Quotient (HQ), among others. In the first equation, the objective is to verify, based on the concentration of the contaminant found in the sample, the average weight of the consumer population, and the food intake rate (how much is consumed in g), whether the intake of the contaminant present in the food exceeds the limits established for consumption by regulatory bodies. In the second equation, the objective is to identify whether there is a risk to human health. In this case, the HQ values were greater than 1, indicating a risk [67]. Of the studies included in this review, only four performed calculations for risk assessment.

In the study by Selvam et al. [31], HQ results for Zn ranged from 0.0045 for adults consuming *Pseudotriacanthus* spp. and *Leiognathus brevirostris* to 0.0562 for children consuming *Atropus atropus*. For Cr, the values were very low, ranging from 0.0001 for adult consumers of *Leiognathus brevirostris* to 0.0009 for children’s consumers of *A. atropus* and *Sufflamen fraenatus*. In this work, both for this and the other metals studied were less than 1. The EDI values in this study ranged from 0.00051 for adult consumers of *Pseudotriacanthus* spp. to 0.05682 for children consuming *A. atropus* for Zn, and 0.00022 to 0.00089 for Cr for the same groups of consumers and species. Both results were considered to be below the limit.

Franco-Fuentes et al. [10] found minimum values of tolerable weekly Zn consumption of 1604 mg week^−1^ for the Ocean whitefish species (*Caulolatilus princeps*) and a maximum of 4015 mg/week for Yellowfin tuna (*Thunnus albacares*). Cr values ranged from 0.025 mg week^−1^ for the Palm ruff species (*Seriolella violacea*) to 0.073 mg/week for Ocean whitefish (*Caulolatilus princeps*). Jiao et al. [44], in their study with squid, also found values of Zn and Cr lower than 1 in the calculation of HQ and results between 0.0050 and 0.0320 for Cr and 0.8214 and 0.8882 for Zn related to the calculation of EDI, also considered below the maximum intake limit. Parolini et al. [30] also identified lower values of HQ < 1 for both metals in their work with mussels.

Shaheen et al. [42] carried out a risk assessment using EDI and THQ and identified a risk (THQ > 1) for the consumption of one of the six fish (*Corica soborn*) species sold in Bangladesh, associated with excess zinc in the samples (THQZn = 1.4). However, this was not observed for chromium. For EDI, the undetermined values were below the tolerable limits for both metals.

It is worth noting that if we consider the highest concentrations found in other studies that did not carry out risk assessment, assuming a scenario of minimum consumption of the species (68 g per week, being consumed only once a week, totaling 48 times of consumption in a year), by young individuals aged 30 years, with an average weight of 78 kg, considering the oral reference dose (RfD) stipulated by USEPA in 2005 of 0.3 mg/kg/day, it was possible to detect values greater than 1, indicating a risk to human health.

As previously described, chromium has carcinogenic potential. In this sense, the assessment of carcinogenic risk (CR) is essential to estimate the possibility of this contaminant causing harm to consumer health. Despite this, few studies have been conducted on such assessments. According to the U.S. According to the Environmental Protection Agency, CR values below 10^−6^ represent low risk, 10^−6^ to 10^−4^ indicate moderate risk, and values above 10^−4^ indicate high risk [67].

Reyes-Márquez et al. [36] found CR values that indicated a moderate to high risk for adult consumers due to the consumption of fish species such as the sea bass *Centropomus undecimalis* (CRCr = 87.7 × 10^−4^) and snapper *Lutjanus griseus* (CRCr = 14 × 10^−4^) from the Tampamachoco coastal lagoon, Gulf of Mexico. Lin et al. [37] also evaluated the cancer risk for Cr contamination in fish samples from the Haikou region, China. The results indicated values compatible with moderate risk in five bivalve species (Razor Clams—1.6 × 10^−4^, White Clams—2.1 × 10^−4^, Fan Shells—1.1 × 10^−4^, oysters —1.3 × 10^−4^, and Blood Clams—2.4 × 10^−4^).

In the CR assessment of Amerizadeh et al. [68], the risk in children was higher than that in adult consumers of the fish species *Cyprinus carpio*, *Rutilus kutum, Rutilus caspicus*, and *Huso huso*. The identified values were 1.4 × 10^−4^ and 7.3 × 10^−4^ for consumers of the mentioned species, adults and children, respectively, from the Golestan region; 2.51 × 10^−4^ and 11.8 × 10^−4^ for adults and children, respectively, from the Mazanderan region, 1.15 × 10^−4^ and 5.61 × 10^−4^ for adults and children, respectively, from the Gilan region, Iran. The results generally indicated a moderate risk of consuming these species, especially for ingesting *C. carpio* by children (CR = 6.78 × 10^−4^). In a study by Shaheen et al. [19], the risk of cancer from consuming six fish species from Bangladesh ranged from 1.0 × 10^−3^ to 9.3 × 10^−4^.

Ray and Vashishth, [38] identified CR values of 4.395 × 10^−4^ for children and 2.513 × 10^−4^ for adults consuming the fish species *Nemipterus japonicus*; 2.816 for adults and 4.93 for children consuming the species *Oreochromis mossambicus*; and 2.697 for adults and 4.757 for children consuming the species Lates calcarifer; indicating that the consumption of the last two species, collected in Tamil Nadu, India, represents a high risk of cancer, especially for children.

The existence of health risks is extremely concerning, especially when it involves an excess of essential elements. Therefore, monitoring studies are important for investigating the contamination status of aquatic environments. These studies should have a defined frequency in regions known to be more contaminated, as well as in control (preserved) areas. They should seek to investigate sentinel species and other highly exploited species, such as those of commercial importance. Surveys should also be standardized so that they can be applied in different regions using the same model, allowing for comparisons.

## 5. Conclusions

The data presented in this article demonstrate the importance of continuously monitoring contaminant levels in large bodies of water and their biospheres, as many studies have shown increasing levels of chromium and zinc not only in places affected by human action but also in environments considered protected, such as the Canary Islands and several estuaries in Europe. This is concerning because these are breeding grounds for many marine species that, as demonstrated in this study, have their reproduction affected by metallic contaminants. The concentrations found in the different studies were generally below the limits and did not pose a risk to consumer health. However, some studies have demonstrated the presence of significant concentrations and indicated both a non-carcinogenic risk related to zinc and a carcinogenic risk associated with Cr. New studies should be encouraged to fill gaps, such as the characterization of the toxicity of these essential elements through seafood consumption; the impact of these elements on marine species during their lifetime; determination of limit concentrations updated by international regulatory institutions, especially for zinc; studies on the influence of abiotic factors on the toxicity and bioavailability of elements in the environment; and those that evaluate the bioaccessibility of these elements in a simulated digestion system when at high concentrations.

## Figures and Tables

**Figure 1 jox-15-00121-f001:**
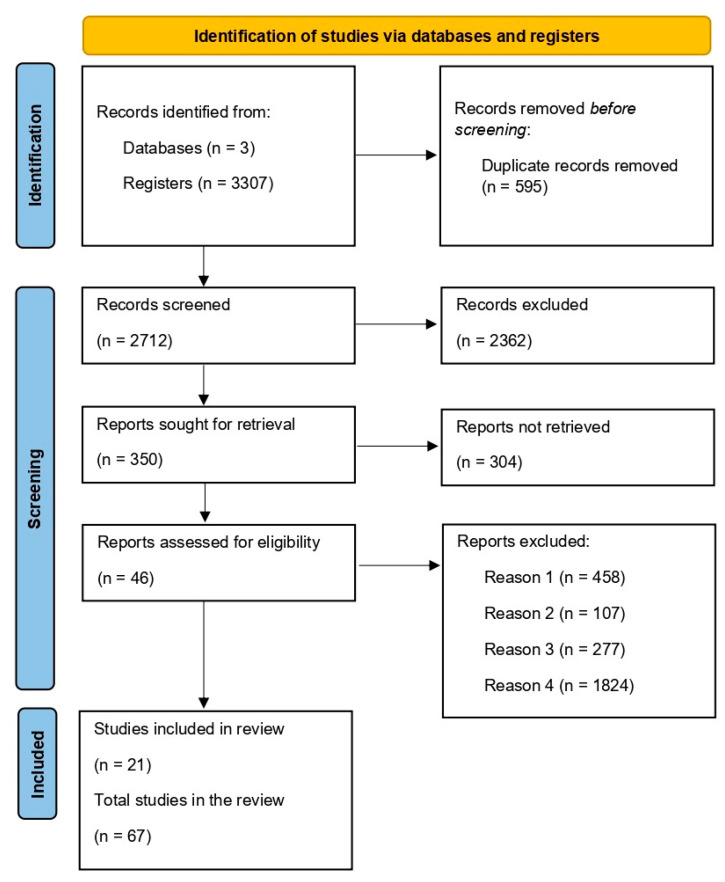
Flow diagram displaying the literature search results (PubMed, Embase, and Web of Science).

**Figure 2 jox-15-00121-f002:**
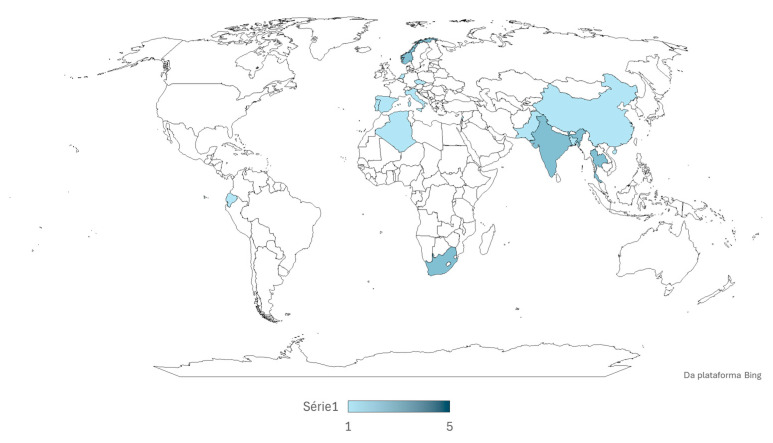
Map identifying all collection spots from the observed articles.

**Figure 3 jox-15-00121-f003:**
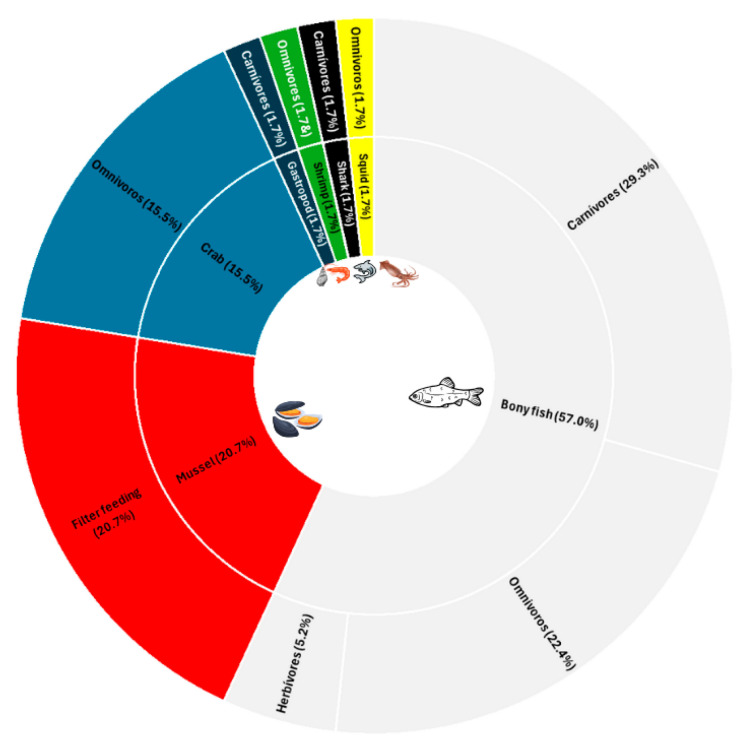
Percentage of marine animals involved in the studies covered in this review and their feeding habits.

**Figure 4 jox-15-00121-f004:**
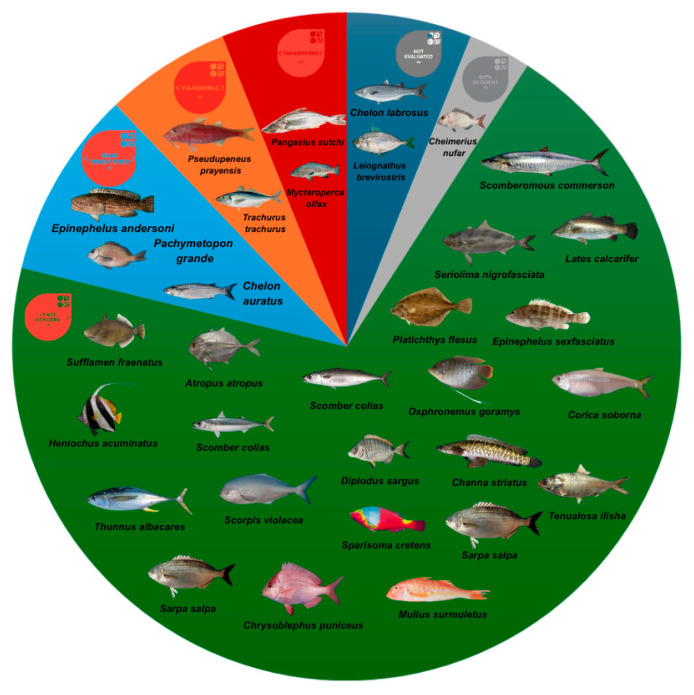
Fish species are covered in this review, and their classification is on the IUCN Red List. The green portion of the graph represents species classified as Least Concern, the light blue portion contains species classified as Near Threatened, the orange portion contains Vulnerable species, the red portion contains Endangered species, the dark blue portion includes unassessed species, and the gray portion contains Data Deficient species.

**Table 1 jox-15-00121-t001:** Risk of bias identified in the main experimental articles selected and grouped by reason.

References	Risk of Bias
Firth et al. [1], Auon et al. [3], Franco-Fuentes [10], Hong et al. [11], Sujitha et al. [12], Dorta et al. [13], Siddiqui and Saher [14], Maulvault et al. [15]	Study location bias—e.g., selection of areas, such as cropland, but without comparative control
Firth et al. [1], Auon et al. [3], Miller et al. [16], Mishra and Mohanty [17], Vutukuru [18], Krumschnabel and Nawaz [19], Wepener et al. [20], Billard and Roubaud [21], Van Pittius et al. [22], Arunkumar et al. [23], Farag et al. [24], Li et al. [25], Mohamed et al. [26], Castro et al. [27], Marinaro et al. [28], McRae et al. [29], Parolini et al. [30], Selvam et al. [31], Lozano-Bilbao et al. [32], Debipersadh et al. [33], Afonso et al. [34], Hanis et al. [35], Reyes-Marquez et al. [36], Lin et al. [37], Ray and Vashishth [38]	Analytical bias—e.g., poor analytical quality control or experimental conditions that cannot be extrapolated to natural conditions, insufficient sampling
Auon et al. [3], Gutiérez-Ravelo et al. [39]	Temporal bias—e.g., lack of sampling that considers the effects of seasonality
Auon et al. [3], Nath and Kumar [40], Ma and Wang [41], Shaheen et al. [42]	Statistical evaluation bias—e.g., lack of statistical details that hinder the full understanding of the evaluation carried out

**Table 2 jox-15-00121-t002:** Concentration (mean or range—minimum and maximum) or mean concentration of Zn and Cr (mg/kg w.w.) in different marine species.

Organism	Species	Zinc	Chromium	Location	References
Fish	*Sufflamen fraenatus*	0.90	0.56	Thoothukudi Coast, India	Selvam et al. [31]
	*Heniochus acuminatus*	1.00	0.46		
	*Pseudotriacanthus* sp.	0.44	0.13		
	*Leiognathus brevirostris*	0.32	0.41		
	*Atropus atropus*	0.28	0.12		
	*Scomber colias*	8.07 ± 3.18	0.20 ± 0.26	Canary Islands	Lozano-Bilbao et al. [32]
	*Scorpis violacea*	10.82 ± 1 1.45	0.13 ± 0.09	Galapagos Islands, Ecuador	Franco-Fuentes et al. [10]
	*Mycteroperca olfax*	15.75 ± 11.20	0.18 ± 0.09		
	*Thunnus albacares*	22.55 ± 7.94	0.16 ± 0.06		
	*Sarpa salpa*	16.3 ± 6.34	0.35 ± 0.68	Canary Islands	Gutiérrez-Ravelo et al. [39]
	*Epinephelus andersoni*	21.0 ± 4.18	0.34 ± 0.15	Durban, South Africa	Debipersadh et al. [33]
	*Chrysoblephus puniceus*	31.63 ± 10.5	-		
	*Cheimerius nufar*	16.37 ± 1.84	1.09 ± 0.32		
	*Pachymetopon grande*	31.07 ± 5.57	1.30 ± 0.11		
	*Trachurus trachurus*	56.71 ± 1.55	1.81 ± 0.47		
	*Scomber colias*	22.06 ± 6.87	0.24 ± 0.43		
	*Diplodus sargus cadenati*	4.51 ± 2.42	0.18 ± 0.15	Canary Islands	Afonso et al. [34]
	*Sparisoma cretens*	2.43 ± 0.37	0.13 ± 0.11		
	*Mullus surmuletus*	5.99–268.36	0.00–2.31	Canary Islands	Dorta et al. [13]
	*Pseudupeneus prayensis*	1.86–14.00	0.00–0.16		
	*Platichthys flesus*	21 ± 4	0.00	Tagus estuary (Portugal), Ebro Delta estuary (Spain), Sacca di Goro estuary (Italy), Western Scheldt estuary (Netherlands) and Solund (Norway).	Maulvault et al. [15]
	*Chelon auratus*	19 ± 1	0.00		
	*Seriolina nigrofasciata*	21.23	6.40		
	*Scomberomous commerson*	24.29	8.09		
	*Lates calcarifer*	22.86	7.41		
	*Epinephelus sexfasciatus*	20.61	7.86		
	*Sarpa salpa*	10.18 ± 2.96	0.12 ± 0.08	Canary Islands	Afonso et al. [43]
	*Chelon labrosus*	3.25 ± 1.80	0.11 ± 0.07		
	*Thunnus* sp.	2.86 to 35.90	0.12 to 1.21	Algeria	Hanis et al. [35]
	*Corica soborna*	392.06 ± 19.22	1.75 ± 0.12	Bangladesh	Shaheen et al. [42]
	*Tenualosa ilisha*	16.42 ± 1.32	6.78 ± 0.40		
Crab	*Ilyoplax frater*	75.44–128	9.35–12.3	Karachi, Pakistan	Siddiqui and Saher [14]
	*Macrophthalmus depressus*	42.60–155	5.65–24.30		
	*Macrophthalmus japonicas*	38.02–85.11	0.37–6.03		
	*Macrophthalmus depressus*	68.8–268.8	-		
	*Opusia indica*	73.94–82.56	7.40–10.94		
	*Austruca sindensis*	2.0–76.9	0.54–0.60		
	*Uca pugilator*	7.00–7.08	<LOD		
	*Uca tangeri*	14.6–81.9	<LOD		
	*Uca annulipes*	10.2–23.8	<LOD		
	*Portunus trituberculatus*	2700	2.3	Lebanon, Mediterranean Sea	Aoun et al. [3]
Mussel	*Mytilus galloprovincialis*	153 ± 23	4.5 ± 0.5	Tagus estuary, Ebro Delta estuary, Sacca di Goro estuary, Western Scheldt estuary, and Solund.	Maulvault et al. [15]
	*Chamelea gallina*	75 ± 3	1.7 ± 0.1		
	*Mytilus edulis*	2384	3.4	Lebanon, Mediterranean Sea	Aoun et al. [3]
	*Patella vulgata*	841	1.22		
	*Choromytilus meridionalis*	15.5 ± 0.32	0.1 ± 0.01	Saldanha Bay, South Africa	Firth et al. [1]
	*Mytilus galloprovincialis*	25.9 ± 0.52	0.2 ± 0.01		
	*Mytilus edulis*	14.87 ± 0.79	0.29 ± 0.01	Norway	Parolini et al. [30]
	*Cerithidea obtusa*	66.0 ± 32	9.3 ± 4.9	Thailand Gulf	Hong et al. [11]
	*Littoraria* sp.	118.2 ± 66	7.1 ± 3.5		
	*Nerita balteata*	69.5 ± 35	13.5 ± 6.7		
	*Ellobium aurisjudae*	150.2 ± 65	9.6 ± 9.9		
	*Isognomon ephippium*	1938.5 ± 835	4.9 ± 2.9		
	*Thais gradata*	657.8 ± 536	20.5 ± 24.2		
Squid	*Loligo chinesi*	15.05	0.542	Shandong Province, China	Jiao et al. [44]
Shrimp	*Litopenaeus vannamei*	12.15–29.08	0.295–1.683	East Midnapore West Bengal, India.	Kumar et al. [45]
Gastropod	*Conus princeps*	9.01 ± 5.28	9.01 ± 5.28	Galapagos Islands	Franco-Fuentes et al. [10]
Limits	Seafood	-	2.0	China	CMLCF [46]
		-	12.0	United States	FDA [47]
		-	2.0	Hong Kong	CFS [48]
		40.0	-	FAO	USEPA [49]
	Water	40.0	0.1	FAO	USEPA [49]

## Data Availability

No new data were created or analyzed in this study. Data sharing is not applicable to this article.

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
