# Peer review of "A Systematic Review on Contamination of Marine Species by Chromium and Zinc: Effects on Animal Health and Risk to Consumer Health"

_jox, 2025, doi:10.3390/jox15040121_

Round 1

Reviewer 1 Report

Comments and Suggestions for Authors

The manuscript is a systematic review analysing the scientific literature on Chromium (Cr) and Zinc (Zn) contamination in marine species. Using the PRISMA protocol, the authors selected 68 studies to assess sources of contamination, health effects on marine animals and risk to consumers of seafood products. The review shows that contamination is widespread, especially in industrialised areas, and that although in many cases levels are below legal limits, some studies indicate a non-carcinogenic (for Zn) and carcinogenic (for Cr) risk to human health. The paper concludes with recommendations for future studies to fill the identified gaps.

Minor revision:

  • The manuscript reported that 323 entries were excluded for “reason 3”. Figure 1 instead shows 23 works excluded for reason 3. Is there an error?
  • In the last section of PRISM graph the authors reported “studies included in the review” and “reports of included in the studies”. What is the mean of these two sentences?
  • The Authors should cite this article: https://doi.org/10.1016/j.cbi.2024.111186
  • Improve the quality of image

Author Response

Comments 1: The manuscript reported that 323 entries were excluded for “reason 3”. Figure 1 instead shows 23 works excluded for reason 3. Is there an error?

Response: Thank you for your comment. A typing error occurred, the values ​​have been corrected.

Comments 2: In the last section of PRISM graph the authors reported “studies included in the review” and “reports of included in the studies”. What is the mean of these two sentences?

Response: Thank you for your comment. The text ​​have been corrected.

Comments 3: The Authors should cite this article: https://doi.org/10.1016/j.cbi.2024.111186

Response: Thank you for your comment. The citation has been added.

Comments 4: Improve the quality of image

Response: Figure quality has been improved.

Reviewer 2 Report

Comments and Suggestions for Authors

This study reviews the contamination of Cr and Zn in aquatic biota and seafood, highlighting their widespread presence, especially in industrial areas. While these elements are essential, excessive exposure through food can pose health risks, including carcinogenic risks associated with Cr. I have some small suggestions.

  1. Abstract: Add the keywords used for literature retrieval.
  2. The introduction provides limited information about Zn. More should be included regarding its sources and its impact on marine ecosystems.
  3. This review has not been registered, which affects transparency.
  4. What were the criteria for excluding studies? The text mentions that studies were carefully evaluated and those that did not present consistent results, discussions, and methods were excluded or contextualized to avoid distortions in the risk estimates.
  5. Materials and Methods: A checklist should be provided.
  6. Discussion: Provide a table with the water quality standards for Cr and Zn, as well as seafood quality standards.
  7. Discussion: Explain how different metals (e.g., Cr(VI) vs. Cr(III) affect bioaccumulation and toxicity.
  8. Discussion: Based on your findings, what are your recommendations for future research?
  9. Discussion: Investigate the impact of Cr and Zn on marine species at different life stages. Typically, the larval stage of crustaceans and mollusks is planktonic, while the adult stage is benthic.
  10. Discussion: Further elaborate on how environmental factors in water influence the chemical species of Cr and Zn.

Author Response

Comment 1: Abstract: Add the keywords used for literature retrieval.

Response: Thank you for your feedback. The changes have been made.

Comment 2: The introduction provides limited information about Zn. More should be included regarding its sources and its impact on marine ecosystems.

Response: Thank you for your comment. We have added the requested information in the introduction

Comment 3: This review has not been registered, which affects transparency.

Response: We register the review in the OSF (https://doi.org/10.17605/OSF.IO/GRXAQ)

Comment 4: What were the criteria for excluding studies? The text mentions that studies were carefully evaluated and those that did not present consistent results, discussions, and methods were excluded or contextualized to avoid distortions in the risk estimates.

Response: Thank you for your comment. The exclusion criteria were summarized in four types: articles that did not study marine species (Reason 1 = 458), studies on the freshwater environment (Reason 2 = 107), studies that dealt only with soil or sediment, without correlation with water or animals (Reason 3 = 277), studies that addressed the results superficially, without clear justifications or results that stood out in relation to other studies that managed to carry out a more in-depth approach (Reason 4 = 1,824). This last reason includes studies that did not present an adequately described methodology that would allow reproducibility by other authors, studies that did not seek to justify or contextualize their results, leaving answers without clarification. These justifications are described in the results topic.

Comment 5: Materials and Methods: A checklist should be provided.

Response: Thank you for your comment. The PRISMA checklist has been added as supplementary material 2.

Comment 6: Discussion: Provide a table with the water quality standards for Cr and Zn, as well as seafood quality standards.

Response: Thank you for your comment, the limits for both metals were added to the results table for easier comparison with obtained results.

Comment 7: Discussion: Explain how different metals (e.g., Cr(VI) vs. Cr(III) affect bioaccumulation and toxicity.

Response: Thank you for your comment, further explanation was added regarding the difference in bioaccumulation between the different metals and metal species.

Comment 8: Discussion: Based on your findings, what are your recommendations for future research?

Response: Thank you for your suggestion. Our opinions and recommendations for future research are outlined throughout the text and emphasized in the conclusion. Some modifications have been made to the conclusion to better emphasize them.

Comment 9: Discussion: Investigate the impact of Cr and Zn on marine species at different life stages. Typically, the larval stage of crustaceans and mollusks is planktonic, while the adult stage is benthic.

Response: Thank you for your suggestion. Unfortunately, information on the toxic effects of Cr and Zn during specific developmental stages in different animals is very scarce. Some information was found about the effects of toxicity in the embryonic stages and was added to the text.

Comment 10: Discussion: Further elaborate on how environmental factors in water influence the chemical species of Cr and Zn.

Response: Thank you for the comment, further explanations on the speciation of both elements were added.

Reviewer 3 Report

Comments and Suggestions for Authors

The manuscript concerns: A systematic review on contamination of marine species by chromium and zinc: effects on animal health and risk to consumer health. It is interesting, but should be improved for publication.

After reading the manuscript, several questions arise:

1. What are the main sources of chromium and zinc contamination in marine environments? Please add.

2. Why are aquatic environments in industrialized regions particularly worrying with respect to these elements? Please add.

3. Is the use of sources [1 and 2] sufficient to support these issues in the text? Please supplement the text with additional literature.

4. Why is it so difficult to establish safe limits for these elements?

5. Can current systems for monitoring fish contamination be improved? How?

6. The first sentence in Materials and Methods is unnecessary. This should be included in the Author Contributions.

7. Figure 5 is illegible. Please correct it.

Comments on the Quality of English Language

It should be corrected

Author Response

 Comment 1: What are the main sources of chromium and zinc contamination in marine environments? Please add.

Response: As described in the introduction and discussion (4.1 Origin and circulation of Cr and Zn in the marine environment), the main source of emissions of these elements into the marine environment is industrial sewage discharged irregularly into this environment. Industrial waste from the steel, fertilizer, paint (mainly antifouling paints), textile and leather tanning industries are the main sources.

Comment 2: Why are aquatic environments in industrialized regions particularly worrying with respect to these elements? Please add.

Response: Thank you for your comment. The major concern regarding industrial areas is related to the irregular emission of industrial sewage. Many of the regions where these metals are quantified in significant concentrations in the marine environment are demonstrably described as areas where sewage is not properly treated and is emitted in its natural state into the water, contributing to the contamination of the site. However, regarding antifouling paints, these are used on vessels and due to the action of salinity, time, pH and other factors, these metals are released into the water over time. This information has been added to the text

Comment 3: Is the use of sources [1 and 2] sufficient to support these issues in the text? Please supplement the text with additional literature.

Response: The references cited above [1 and 2] support the statements in the text, however, two more citations have been added that also support the text [3,4, and 5].

Comment 4: Why is it so difficult to establish safe limits for these elements?

Response: Thank you for your comment. Since these elements are essential to human and animal health, and perform fundamental functions, there is a very subtle threshold between essential and toxic concentrations, depending on the organism, such as aspects of animal/human metabolism, environmental conditions that may interfere, and also the chemical species absorbed and its interaction with the organism. Comparatively, Cr has chemical species that have been well studied in toxicological tests, especially hexavalent chromium, and consequently has better defined limits, with both the limits in fish and the concentration required for daily consumption to achieve beneficial effects being known (Cr (III) of 1.5 mg per day and 0.003 mg per day for Cr (VI)). On the other hand, zinc toxicity is less studied, its daily consumption range is higher (8 mg per day) and according to the studies, the concentrations that indicate toxic effects are quite variable, depending on the factors mentioned.

Comment 5: Can current systems for monitoring fish contamination be improved? How?

Response: Thank you for your comment. Yes, monitoring studies should have a set frequency in regions known to be more contaminated, but also in control (preserved) areas. They should seek to investigate sentinel species, but also other species that are highly exploited, such as those of commercial importance, and the research should be standardized so that it can be applied in different regions using the same model, allowing for comparisons.

Comment 6: The first sentence in Materials and Methods is unnecessary. This should be included in the Author Contributions.

Response: Thank you for your comment. We have modified this topic to contain only relevant information about article inclusion and exclusion criteria, as well as the step-by-step process for carrying out the review.

Comment 7: Figure 5 is illegible. Please correct it.

Response: Thank you for your feedback. Image quality has been improved.

Round 2

Reviewer 2 Report

Comments and Suggestions for Authors

The authors have addressed all of my previous comments.